# Use of an Argentine Ant, *Linepithema humile*, Semiochemical to Deliver an Acute Toxicant

**DOI:** 10.3390/insects9040171

**Published:** 2018-11-23

**Authors:** Benjamin M. Gochnour, Daniel R. Suiter, Jerry W. Davis, Qingguo Huang

**Affiliations:** 1Department of Entomology, University of Georgia, 120 Cedar Street, Athens, GA 30602, USA; bmg1110@gmail.com; 2Department of Entomology, University of Georgia Griffin Camus, 1109 Experiment Street, Griffin, GA 30223, USA; 3Experimental Statistics, College of Agricultural and Environmental Sciences, University of Georgia, 1109 Experiment Street, Griffin, GA 30223, USA; jwd@uga.edu; 4Department of Crop and Soil Sciences, University of Georgia Griffin Campus, 1109 Experiment Street, Griffin, GA 30223, USA; qhuang@uga.edu

**Keywords:** Argentine ant, *Linepithema humile*, Formicidae, semiochemical, ant control, fipronil

## Abstract

The Argentine ant, *Linepithema humile*, is an invasive nuisance, agricultural, and ecological pest from South America. In the United States, its primary distribution is in California and the Southeast. The structural pest control industry responds to property owner complaints when this ant’s populations become problematic and a persistent nuisance. Actions taken to control Argentine ants in the urban and suburban environment are typically complaint-driven, and often involve the application of insecticide sprays applied to the outdoor environment by professional pest managers. In California, and elsewhere, spray treatments of various residual insecticides by property owners and pest management professionals has resulted in significant runoff and in subsequent surface water contamination. As a result, an immediate need exists to develop alternative methods of ant control targeted at reducing environmental contamination. The purpose of this study was to investigate the potential for the development of an alternative method of toxicant delivery focused on the Argentine ant’s behavior modifying cuticular chemistry. In short, methanol and hexane washes of Argentine ant pupae applied to paper dummies were handled significantly more by worker ants than the paper dummies that did not contain the solvent extracts. Additionally, paper wicks soaked in a methylene chloride wash from Argentine ant cadavers, air dried, and then treated with fipronil, were removed by worker ants and placed on a midden pile at the same rate (≈86% to 99% removal at 1 h) as untreated and fipronil-treated ant cadavers. The paper wicks that did not contain the methylene chloride extract were ignored by the worker ants. After three days, the mortality of the ants exposed to the fipronil-treated wicks or the ant cadavers were dose-related. In conclusion, our study suggests that there is potential for the use of ant semiochemicals for the delivery of acute toxicants.

## 1. Introduction

The Argentine ant, *Linepithema humile* (Mayr), is a major nuisance pest, and in the United States, it is found mainly in southern California and the Southeast [1]. It commonly manifests as a supercolony, and is able to dominate entire landscapes and become highly invasive in human altered habitats [2]. The supercolony contains many non-competing, reproductive queens that drive the abundance of this species, where it occurs [3].

Contact insecticides are frequently used to control pest ant species such as the Argentine ant [4]. Although research has shown fipronil baits as being effective in the laboratory, Argentine ant control methods utilize mostly sprays in and around structures [5]. Fipronil (IUPAC: 5-amino-1-[2,6-dichloro-4-(trifluoromethyl)phenyl]-4-(trifluoromethylsulfinyl)pyrazole-3-carbonitrile), as a bait and contact insecticide, is able to achieve a high ant mortality at relatively low concentrations, because it is readily transferred among worker ants through physical contact prior to succumbing [6]. In baits, its toxicity is delayed long enough for it to spread to critical colony members, such as queens [7,8].

Delayed toxicity is a critical factor in the effectiveness of insecticidal agents against the Argentine ant [4,9,10]. In a laboratory experiment, Wiltz et al. [4] conducted a series of tests to evaluate the effectiveness of four insecticides at killing Argentine ants. The workers were topically treated with the chemicals, and impairment and median lethal times were recorded. The time to mobility impairment was slowest in fipronil treated colonies, although mortality was also the highest in the fipronil treated colonies, demonstrating fipronil’s ability to be spread effectively among the colony. Choe and Rust [6] tested eight insecticides for their ability to be spread by contact. After exposure to insecticide treated sand for one minute, the ants were placed into a colony of untreated ants. Only fipronil, at rates of 0.002% and 0.004%, resulted in an adequate contact transfer and the subsequent high mortality rates within four days. The mortality rates for the remaining insecticides were not significantly different from the control. A second experiment using fipronil treated ant cadavers showed that necrophoresis played an important role in the contact transfer of the insecticide. Fipronil-treated cadavers placed closer to the nest resulted in a higher mortality compared with the cadavers placed 30 cm from the nest.

Vega and Rust [11] marked foraging Argentine ants with a fluorescent brightener (FB28) to determine the origin of resurgence after treatment with fipronil baits and sprays. Baits containing 0.0001% fipronil and sprays formulated with 0.06% fipronil achieved a significant reduction in Argentine ant workers after four weeks. The percentage of marked ants in these treatments decreased throughout the experiment as a result of immigration from the surrounding areas, suggesting that in order to effectively control Argentine ants, a much larger treatment area may be necessary. Klotz et al. [12] reported the effective control of Argentine ants around homes using a combination of baits and a barrier spray treatment. While both methods were effective at controlling the ants, baiting required several reapplications in order to maintain adequate control, while a single application of the spray was equally effective.

Environmental concerns over the use of liquid spray insecticides for ant control [13,14,15,16,17] have led to the investigation of alternative methods for Argentine ant control. Essential oils from plants are a deterrent to ants [18,19,20]. The use of Argentine ant trail pheromone mixed with a liquid bait enhanced recruitment to and consumption of the bait [21], and trail pheromone mixed with a fipronil spray enhanced Argentine ant mortality compared with a fipronil spray treatment alone [22]. The efficacy of the spray pheromone is dependent on the application method and the ant trail density, with point source applications being more effective, and areas of heavy Argentine ant establishment being less affected [23,24,25]. Efforts have also been made to reduce the amount of fipronil needed to maintain acceptable control of Argentine ants by using granular formulations, instead of sprays, that are not as easily transferred to water sources [26]. The use of novel, species specific attractants, such as fipronil-treated termites, for the control of the Asian needle ant, *Brachyponera chinensis* Emery, and Argentine ants, allows for the targeted application of the insecticide, while reducing the amount of insecticide introduced into the environment [27].

Semiochemical-based retrieval behaviors by the Argentine ant may present a partial solution to the issue of insecticide volume and application method. Evidence suggests that ant retrieval behavior, where the ants are signaled to move something into or away from their nest, can be requisitioned by other insects and plants in order to aid in egg and seed dispersal [28,29]. Elaiosomes on plant seeds and their mimics, the capitula on stick insects’ eggs, can cause certain species of ants to pick up and move these objects to a desired location for incubation. The stick insect’s egg capitulum has a large lipid component believed to be an incentive for retrieval, similar to the large lipid component of elaiosomes found on some plant seeds [30].

The chemistry underlying necrophoresis, the transport of dead individuals away from the colony, was explored by Choe et al. [31] in Argentine ants, and it was revealed that the cuticular chemicals eliciting the response are always present whether ants are alive or dead. In live ants, the signal is masked by two highly volatile compounds, dolichodial and iridomyrmecin, thus preventing conspecific necrophoretic behavior toward live workers. When an ant dies, the volatile compounds evaporate or degrade quickly, revealing the masked cuticular odor and eliciting the associated retrieval behavior. Necrophoresis played an important role in the spread of fipronil among Argentine ant workers when the worker cadavers treated with the insecticide were placed near the nest [6].

Fipronil sprays are effective at controlling pest populations of Argentine ants, but sprays have ecological drawbacks. The need for alternative control methods is warranted. The intent of our study was to explore the possibility of using an Argentine ant semiochemical as a means of delivering small quantities of a contact insecticide. Extracts containing a necrophoretic signal molecule from the Argentine ant cuticle were tested in combination with a fipronil suspension, in order to assess the viability of this novel approach in ant management.

## 2. Materials and Methods

### 2.1. Study Organism

Leaf litter debris containing Argentine ant nests (workers, queens, and all stages of brood) were collected from the field on the University of Georgia Griffin campus in Griffin, GA in June and July 2017. The ants and debris were kept in PTFE fluoropolymer-lined (Insect-a-Slip, BioQuip Products, Inc., Rancho Dominguez, CA, USA) rearing bins (56 × 43 × 13 cm) and provided test tubes containing water or 20% sucrose water stoppered with a cotton ball. The PTFE fluoropolymer kept the ants from escaping. Freshly killed house crickets, *Acheta domestica*, were provided ad libitum as a protein source. The nesting chambers consisted of a 90 × 18 mm Petri dish filled three quarters of the way with Castone (Dentsply Sirona, York, PA, USA). Castone is water absorbent and increases the humidity inside the nest chamber when wetted. Holes were drilled in either side of the Petri dish to serve as entrances to the nesting chamber. The ants moved as the leaf litter dried into the moistened, Castone-filled dishes, allowing for the easy collection of workers, queens, and brood.

### 2.2. Experimental Colonies

Experimental colonies were created by removing a nesting chamber, containing multiple queens, thousands of workers, and grams of brood, from a rearing bin, and placing it in the corner of a Pyrex glass tray (24.5 × 20 × 7.5 cm; Asahi Glass Company, Tokyo, Japan) with PTFE fluoropolymer on the inner walls so as to prevent any ants escaping (Figure 1). The colonies were provided 20% sucrose water and water, as described above, and were allowed to acclimate for 24 h before being used in the experiment.

### 2.3. Retrieval of Intact Pupae

Nesting chambers containing queens, workers, and brood were removed from the rearing bins and were placed in a PTFE fluoropolymer-lined plastic container (19.5 × 14 × 10 cm), and the ants were immobilized with carbon dioxide gas. Pupae were removed under a microscope using a small paintbrush. A single pupa was placed in the center of the open area of an experimental colony (Figure 1d), and the time required for the worker ants to retrieve the pupa was recorded. Retrieval was defined as a worker picking up and bringing the pupa to a nest entrance. Ten experimental colonies were assayed three times for a total of 30 replicates. The experiment was repeated once per colony using pupae killed by freezing, in order to determine worker ant behavior in response to lifeless pupae.

### 2.4. Activity of Pupal Extracts

The cuticular extracts from the ant pupae were obtained by placing two pupae onto a piece of filter paper (2 × 3 mm; Whatman #5) and applying 10 µL of hexane or methanol in single drops directly onto the pupae. The pupae were discarded and the solvent was allowed to evaporate for one hour before bioassay of the filter paper.

A 30 mm diameter white paper disc was placed under the glass tray of an experimental colony in the center of the open area in front of the nesting chamber (Figure 1d). A piece of filter paper, with or without a cuticular extract, was placed in the center of the circle onto the glass bottom of a tray containing an experimental colony. The number of ants surrounding and on top of the filter paper and inside the circle’s border were recorded every minute for 20 min. The treatments consisted of a no item control (nothing in the circle), an untreated filter paper, a methanol treated filter paper, and a filter paper containing hexane- or methanol-extracted cuticular compounds from two Argentine ant pupae. Each treatment was replicated five times.

### 2.5. Activity of Fipronil-Treated Ant Cadaver Extracts

Ant cadavers were obtained by placing live workers into a standard freezer for 24 h, removing them, and allowing them to air dry for one hour. About 600 dry cadavers were placed into a small beaker, and 3 mL of methylene chloride was added and gently swirled for two minutes. The ants were removed and the solvent was decanted into a clean beaker containing 200 cylindrical, paper wicks (MiTeGen, Ithaca, NY, USA, extra fine wicks; 3 × <1 mm). The solvent was allowed to evaporate and the wicks were allowed to air dry for an additional hour. The process was repeated, without ants, in order to obtain methylene chloride treated wicks (negative control). Fipronil (0.001%, 0.01%, and 0.10%; Termidor SC, BASF Corp, Research Triangle Park, NC, USA) was applied to one-hour old ant cadavers, methylene chloride treated wicks, and wicks containing the cadaver extract with a coarse mist (≈10 mL), from a hand-held spray bottle, so as to lightly cover all of the treated items. The fipronil-treated items were allowed to dry for one hour, and then 30 were gently placed into a small, experimental colony. Each treatment was replicated six times.

The experimental colonies for this experiment were created by adding ≈300 live worker ants (estimated gravimetrically) into a PTFE fluoropolymer-lined Pyrex glass tray containing an empty, moistened nesting chamber (Figure 2). The ants were allowed to acclimate for 24 h before the experiments were initiated. The experiments were initiated by placing 30 treated items (ants or wicks) onto a flat, plastic platform fashioned from one-third of a weigh boat (Figure 2b), and the platform was then placed in the center of the open space in the glass dish.

### 2.6. Response Variables

The number of items removed from the plastic platform were recorded after one, two, and 24 h so as to obtain removal counts. Live and dead ants in each experimental colony were removed after 72 h and were counted. Percent mortality was determined by dividing the number of dead ants by the number of dead ants plus the number of alive ants, and multiplying this product by 100.

### 2.7. Statistical Analysis

#### 2.7.1. Activity of Pupal Extracts

The results from the activity experiment were count data. The number of ants inside the border of the 30 mm circle at each minute, for the sum of 20 min, were analyzed using a negative binomial distribution by PROC GLIMMIX, with colony as a random effect [32]. Differences in least square means were determined by pairwise *t*-tests.

#### 2.7.2. Activity of Fipronil-Treated Ant Cadaver Extracts

The number of items removed by the live ants at one, two, and 24 h were analyzed by one-way analysis of variance (ANOVA) using PROC GLIMMIX [32]. The removal counts were modeled using a Poisson distribution with colony as a random effect. The mortality was modeled using a binomial distribution with replicate as a random effect, and the mortality was analyzed by one-way analysis of variance with events by trials syntax using PROC GLIMMIX [32].

## 3. Results

### 3.1. Pupa Retrieval Time

All of the pupae were picked up by live worker ants and were moved into the Castone nest. The worker ants retrieved pupae at a mean time of 289.3 s (median = 246.5, range 56–688, *n* = 30). In addition, all 10 of the dead pupae were retrieved.

### 3.2. Activity of Pupal Extracts

The treatments were separated into three statistically distinct groups (*F* = 49.2; df = 4.16; *p* < 0.0001; Figure 3). The no item control treatment reflected the random occurrence of workers in the circular activity area without an introduced stimulus. This resulted in little activity, averaging 2.5 ants over the duration of the experiment. The addition of an inert item, with or without exposure to just the solvent, increased the ant response by 14-(no-solvent blanks) to 17-fold (methanol only). There was no difference in ant response to wicks (no solvent) and to wicks treated with methanol that were allowed to dry. The treatments containing either of the pupal extracts (hexane or methanol) were not significantly different, and elicited an activity that was more than three times that of any treatment not containing a pupal extract. The methanol extract elicited the greatest ant activity, with an average of 169.4 ants counted in the activity area, followed closely by the hexane extract.

### 3.3. Activity of Fipronil-Treated Ant Cadaver Extracts

The ant cadavers and the wicks containing the methylene chloride extract from the ant cadavers were removed in significantly greater numbers (e.g., 25.9 to 29.6, or 86.3% to 98.7%, of the 30 items) than the wicks containing no ant cadaver odors (e.g., 0.7 to 1.7, or 2.3% to 5.7%, of the 30 paper wicks) (Table 1, hour 1; *F* = 23.0; df = 9,45; *p* < 0.0001). Furthermore, the fipronil-treated cadavers and the wicks containing the extract from ant cadavers were removed at the same rate as the ant cadavers that were not treated with fipronil, suggesting that (a) the bioassay duplicated the retrieval response by worker ants toward ant cadavers, and that (b) the fipronil insecticide formulation was not a deterrent at the concentrations tested. Lastly, the wicks that did not contain ant cadaver odors were ignored by the ants, and were not removed (0.7 to 1.7 wicks removed after one hour). After two and 24 h, the pattern of removal had not changed appreciably, either in the number removed or in statistical significance. After 24 h, retrieval in the treatments containing ant cadavers, or in the wicks with an ant cadaver extract, ranged from 90% to 100%, while retrieval in the treatments with wicks containing no extract ranged from 16% to 45.7%.

Regardless of the presence of ant cadaver odor, mortality of the ants generally increased with fipronil concentration, and was (a) 19.3% to 84.9% in fipronil treated ant cadavers; (b) 14.0% to 43.9% in paper wicks treated with the ant cadaver extract and fipronil; and (c) 18.3% to 51.9% in the fipronil-treated wicks (no cadaver extract) (Table 2; *F* = 425.7; df = 9,50; *p* < 0.0001). In seven of the nine fipronil treatments, mortality was significantly greater than in ants exposed to ant cadavers that were not treated with fipronil (untreated control treatment). Mortality in the untreated ants (control treatment) was 15.1%. In general, regardless of fipronil concentration, the descending order of treatment activity was as follows: fipronil-treated ant cadavers > fipronil-treated wicks (no cadaver odor) > fipronil-treated wicks (with cadaver odor). The mortality of the ants exposed to fipronil treated cadavers was more consistent and had a wider range (range = 65.6%) than the mortality of ants exposed to wicks treated with the ant cadaver extract and fipronil (range = 29.9%) and the wicks treated with fipronil but not the cadaver extract (range = 33.6%).

The lowest concentration of fipronil tested (0.001%) was ineffective as a contact toxicant. The mortality in 0.001% of the fipronil treatment groups, where the cadaver odors were present (14.0% or 19.3%), was similar to the mortality in the untreated control group (no fipronil) (15.1%), and in the treatment where the wicks were treated with fipronil, but not the cadaver extract (18.3%).

The exposure of ants to 0.01% fipronil-treated cadavers was significantly more active (58.9% mortality) than the fipronil-treated paper wicks containing the cadaver extract (13.5% mortality) or the paper wicks treated with fipronil only (25.9% mortality). Possibly owing to the nature of the paper wicks, the mortality was significantly greater in the ants exposed to the wicks treated with only the insecticide (25.9% mortality) than in the wicks treated with the cadaver extract and 0.01% fipronil (13.5% mortality, which was not significantly different from the untreated control).

The mortality in the 0.10% fipronil series followed the same general pattern as in the 0.01% treatment series—that is, the mortality in the fipronil-treated ant cadaver group (84.9% mortality) was significantly greater than in either group where the wicks were treated with fipronil and the cadaver extract (43.9% mortality), or were not treated with the cadaver extract (51.9% mortality). As was the case in the 0.01% fipronil series, the mortality was significantly greater in ants exposed to the wicks treated with only the insecticide (51.9% mortality), than in the wicks treated with the cadaver extract and 0.10% fipronil (43.9% mortality).

## 4. Discussion

### 4.1. Pupa Retrieval Time

The pupa retrieval experiment demonstrated that Argentine ants have a strong tendency to retrieve immatures; all of the pupae (*n* = 30) were retrieved in less than 12 min (mean and median retrieval times of 4.1 and 4.8 min). The dead pupae were also returned to the nest. In the red imported fire ant, *Solenopsis invicta* Buren, cold-killed pupae <21 h old were retrieved by the workers [33]. Walsh and Tschinkel [33] suggest that the “persistence of the signal for such long periods after death indicates either that the cue is present in large quantities, or is extremely potent and/or stable; also, auditory brood communication can be ruled out”. A number of published studies have demonstrated that compounds on the surface of ant eggs, larvae, and workers elicit specific behavioral responses from nestmates, including the destruction of foreign eggs, the recognition and care for larvae, and the removal of nestmate cadavers from the nest [31,34,35,36,37,38,39,40].

### 4.2. Activity of Pupal Extracts

The results from the activity measuring experiment provide basic, preliminary insight into the chemical ecology of pupa retrieval by ant workers. The no item control demonstrated a baseline activity that would be expected from the random movement of foragers in the arena environment. The blank filter paper and methanol-treated filter paper increased the activity of the foragers in the sample area, providing evidence for worker ant interest in new, inert objects in their foraging area. The ant response to the methanol-treated, and dried, filter paper further demonstrated that the methanol had no impact on response, as interest in it was not significantly different from the blank control. When the solvent-based cuticular extracts were added to the filter paper, the activity in the sampling area was enhanced four- (hexane) or five (methanol)-fold. Further evidence in fire ants suggests that their pupal skins contain a contact, brood recognition pheromone(s) that elicits retrieval behavior in worker ants. In their study, ant pupae rinsed with small aliquots of hexane, methanol, ether, or benzene lost their attractiveness to the worker ants [33].

The goal of our study was to contribute to the development of alternative techniques in the delivery of traditional insecticides when the target pest is ants. Enticing ants to carry insecticide-treated (in our case an acute, contact insecticide) items deep into central nest sites may prove useful. The chemicals used to entice ants to pick up an otherwise inert object (paper wick) treated with insecticide need not be restricted to behavior-modifying compounds (i.e., pheromones). To our knowledge, the idea was first proposed in 1975 [41]. In this study, commercially-available triolein and diolein applied to paper discs at 1% in an organic solvent enticed red imported fire ant workers to retrieve the discs and to take them into the nest. The behavior was likely a food response and not, as the authors stated, the identification of a brood pheromone [42]. Wiltz et al. [43] applied triolein and fipronil to inert corn cob grits and enticed red imported fire ants to pick up and handle the material, resulting in enhanced ant mortality. In the field, the grits treated with triolein or triolein and fipronil were removed at a significantly higher rate (52% to 74% removal) than the untreated grits (26% removal). Although individual compounds were not identified, otherwise inert corn cob granules coated with a crude hexane extract of whole fire ant larvae were collected from outside the nest; brought into the nest; and placed next to the intact, live immatures [44]. The extract-containing granules were arguably treated as if they were intact larvae. A viable explanation for this behavior was never reached, but the response may have, again, been a food response. The authors suggest that the response was the result of the presence of a brood-recognition pheromone, but others disagreed [33]. Regardless of the reason, the granules were moved into the central nest sites where they were placed next to the larvae. Had these granules been treated with insecticide, the results may have been catastrophic for the ant colonies.

Elaiosomes are nutrient rich food bodies attached to plant seeds (taken together, they are referred to as a diaspore) that entice numerous ant species to pick up and disperse seeds while utilizing the elaiosome as a food source. Inert granules coated with the polar lipid fraction from elaiosomes (methanol solvent fraction) were as attractive to the ants as intact elaiosomes; furthermore, oleic acid, 1,2-diolein, and triolein were particularly attractive to the ants, and were also collected as frequently as the intact elaiosomes [30,45]. Interestingly, some tropical species of stick insects (Phasmatidae) produce eggs that mimic plant diaspores in appearance and, perhaps, elaiosome chemistry. Attached to the insect egg are small knobs, called capitula, that are attractive to ants because they contain a substantial lipid content [28]. Ants collect the stick insect eggs and return them to their nests, much like they collect and return plant diaspores. In what is arguably a case of convergent evolution, the capitula is an elaiosome mimic that aids in the protection of eggs from Hymenopteran parasitoids [28]. The eggs with capitula and the seeds with elaiosomes were removed by the ants at the same rate and at a significantly faster rate than the eggs without the capitula and the seeds without elaiosomes, respectively [28,29]. This entire phenomenon is driven by the lipid food content of the capitula.

Other studies have involved the movement of insecticide-treated, live prey (food) into the pest ant’s central nest site, resulting in major reductions in the pest’s numbers [46,47]. In both laboratory and field trials, the Asian needle ant and the Argentine ant were enticed to collect and feed upon live termites exposed to small amounts of fipronil. The termites were collected and brought into the target ant’s nest site, and were dismembered and consumed. The introduction of an insecticide-contaminated item, in this case live prey, into the central nest site resulted in the complete contamination of this central core of the ant’s colony with an acute toxicant, resulting in major reductions, if not the outright elimination, of the target pest ant.

Ant behavior-modifying semiochemicals, including the one we used in our experiments, are obvious targets for use as a component in an alternative model for insecticide delivery to ants. To our knowledge, the chemical structure of a brood recognition pheromone (for pupae) in ants has not been made, however, there is strong evidence that pupal recognition pheromones exist and that they elicit retrieval into the nest from areas external to the nest [33,48]. Walsh and Tschinkel [33] were clearly able to remove this chemical(s), from the outside of the fire ant pupae, with washes in various organic solvents. The washed pupae were no longer attractive to the workers. When not washed, the worker ants engaged in strong and consistent retrieval behavior when the pupae were placed outside the nest. In our study, the Argentine ant workers retrieved 100% of the live (*n* = 30) and freeze-killed (*n* = 10) conspecific pupae when placed on the floor of the glass arena outside the nest.

Although our preliminary experiments showed that Argentine ants do indeed respond to both hexane- and methanol-washes of ant pupae, the ants did not mimic the pupa retrieval behavior and did not move the treated papers into the nest. Perhaps if the assays were allowed to proceed for a longer period, they may have. Clearly, this aspect of our study warrants further investigation. Using the ant’s retrieval behavior to deliver insecticide treated objects deep into the heart of the nest might reduce the amount of insecticide needed for appreciable levels of control.

### 4.3. Activity of Fipronil-Treated Ant Cadaver Extracts

The removal experiment revealed the strong handling response for all of the experimental treatment groups that were composed of an ant cadaver, or the methylene chloride extract of the ant cadavers. The removal rate of the ant cadavers, the ant cadavers treated with fipronil, and the fipronil-treated wicks coated with the methylene chloride cuticular extract from the dead ants were not statistically different, demonstrating that the extract elicited a retrieval response similar to that of the recently killed ants, and that the insecticide formulation was not a deterrent. Although preliminary, this system has the potential to allow for a targeted application of insecticides, thereby helping to reduce the potential for insecticide run-off and ground water contamination [13,14,17,49,50].

Reconciling the removal data (Table 1) with its associated mortality (Table 2) was inconsistent. The removal was clearly associated with the presence of the extracted cuticular compounds, and the mortality generally increased with the increasing rates of fipronil, a trend found in previous studies [6]. The ant cadavers treated with fipronil generated the greatest mortality at each fipronil rate, possibly because of a difference in the amount of the fipronil available for contact toxicity by worker ants compared with the treatments with wicks. The ant cadavers likely did not absorb liquid like the wicks, and so more of the insecticide may have remained on the surface of the ant cuticle, resulting in greater bioavailability.

The near complete removal but lower mortality in the extract and wick treatments when compared with the fipronil only wicks (no extract), which had low removal rates yet a higher mortality at each fipronil concentration tested, confounded the relationship between the wick removal and the mortality of the ants handling them. This discrepancy may be explained by the acute activity of fipronil to worker ants. In the activity measure experiment (Figure 3, blank control), the mere presence of a foreign object was enough to elicit a 14-fold increase in the activity around and on the object. The interaction with wicks (touching and crawling on and around), even without removal, may have provided ample opportunity for the ants to acquire lethal doses of fipronil via contact, especially at the high rates tested (0.01% and 0.10%).

Our study’s aim was to entice Argentine ants to pick up and handle an insecticide-treated, inert item (a piece of paper) coated with a methylene chloride extract containing a cuticular compound, from dead ants, which entices nestmates to pick up and handle treated items [31]. Although the ants removed the methylene chloride-treated wicks at the same rate as the cadavers, the application of fipronil to the system was only mildly successful. The novelty of this insecticide delivery method, and its potential to reduce environmental contamination, warrants further exploration. Clearly, other candidate chemicals, both food- and behavior-based, are available for the co-treatment of an inert carrier (e.g., paper wicks) with an acute toxicant, which might be delivered deep into the inner core of an ant nest, thereby eliminating the colony from within. Reducing the amount of insecticide needed to control Argentine ants would have a significant and positive ecological impact, and may lead to new control methods for other pest ant species.

## Figures and Tables

**Figure 1 insects-09-00171-f001:**
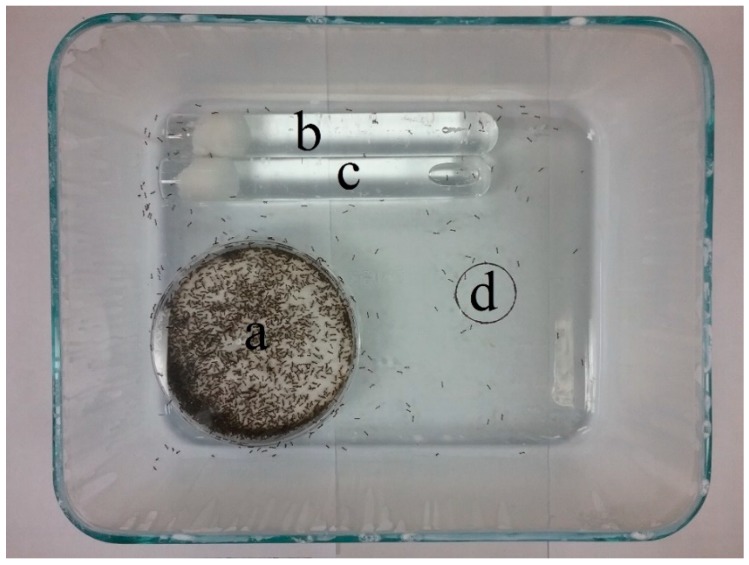
Experimental colonies consisted of a nesting chamber (**a**) filled with multiple queens, thousands of workers, and grams of brood, and tubes filled with water or 20% sucrose water (**b**) and (**c**). Intact live or dead pupae (Section 2.3) or paper wicks containing a methanol or hexane extract (Section 2.4) were placed in the center of a 30 mm diameter circle (**d**).

**Figure 2 insects-09-00171-f002:**
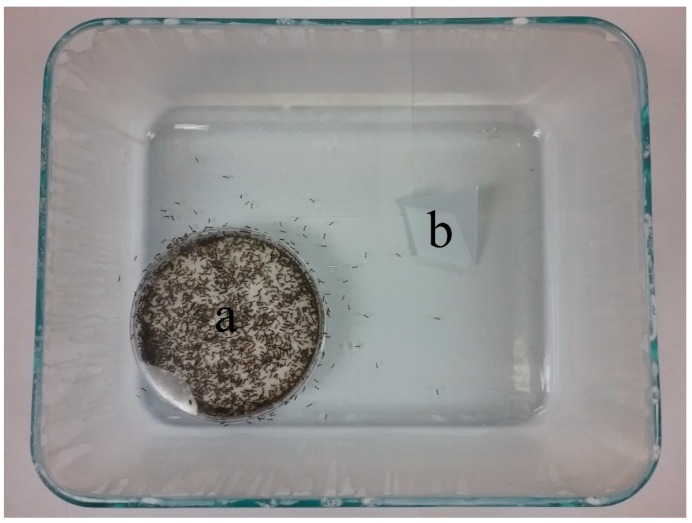
Test arena for item removal and mortality experiments. Nest chamber (**a**) containing ≈300 worker ants was provided, and 30 items (ants or paper wicks) were placed onto the floor of the arena on a plastic platform (**b**).

**Figure 3 insects-09-00171-f003:**
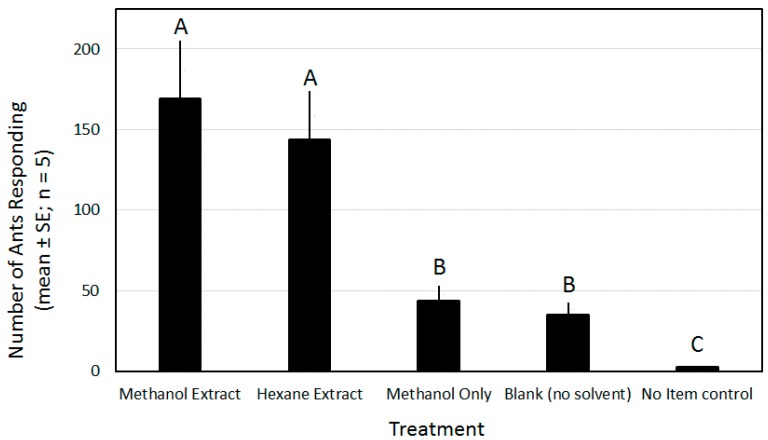
Number of Argentine ant workers (mean ± standard error (SE); *n* = 5 replicates per mean) surrounding a paper wick, with or without a pupal extract, placed in the center of a 30 mm diameter circle on the floor of a small, glass test arena. Means followed by the same letter are not significantly different [32].

**Table 1 insects-09-00171-t001:** Number (mean ± standard error (SE); *n* = 6 replicates per mean) of fipronil-treated and untreated Argentine ant cadavers and paper wicks (*n* = 30) removed at one, two, and 24 h by Argentine ant workers. Means within a column, and followed by the same letter, are not significantly different [32].

Treatment	Hour 1	Hour 2	Hour 24
Ant Cadavers	26.1 ± 2.6 a	27.8 ± 2.2 a	30.0 ± 2.2 a
0.001% fipronil + cadavers	26.8 ± 2.6 a	30.0 ± 2.3 a	30.0 ± 2.2 a
0.01% fipronil + cadavers	26.8 ± 2.6 a	30.0 ± 2.3 a	30.0 ± 2.2 a
0.10% fipronil + cadavers	25.9 ± 2.5 a	27.8 ± 2.2 a	30.0 ± 2.2 a
0.001% fipronil + cadaver extract + wicks	27.7 ± 2.7 a	30.0 ± 2.3 a	30.0 ± 2.2 a
0.01% fipronil + cadaver extract + wicks	29.6 ± 2.8 a	30.0 ± 2.3 a	30.0 ± 2.2 a
0.10% fipronil + cadaver extract + wicks	25.9 ± 2.5 a	26.5 ± 2.2 a	27.0 ± 2.1 a
0.001% fipronil + wicks	0.7 ± 0.3 b	0.8 ± 0.4 c	7.0 ± 1.1 c
0.01% fipronil + wicks	1.5 ± 0.5 b	2.0 ± 0.6 bc	4.8 ± 0.9 c
0.10% fipronil + wicks	1.7 ± 0.5 b	2.7 ± 0.7 b	13.7 ± 1.5 b

**Table 2 insects-09-00171-t002:** Mortality of Argentine ants three days following exposure to insecticide-treated and untreated ant cadavers and paper wicks containing a methylene chloride extract of ant cadavers. Means (based on *n* = 6 replications) followed by the same letter are not significantly different [32].

Treatment	Percent Mortality (Mean ± SE)
Ant cadavers	15.1 ± 4.3 g
0.001% fipronil + cadavers	19.3 ± 1.2 f
0.01% fipronil + cadavers	58.9 ± 7.3 b
0.10% fipronil + cadavers	84.9 ± 4 a
0.001% fipronil + cadaver extract + wicks	14.0 ± 2.1 g
0.01% fipronil + cadaver extract + wicks	13.5 ± 1.5 g
0.10% fipronil + cadaver extract + wicks	43.9 ± 7.9 d
0.001% fipronil + wicks	18.3 ± 6.2 f
0.01% fipronil + wicks	25.9 ± 5.1 e
0.10% fipronil + wicks	51.9 ± 5.9 c

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
