# Peer review of "Use of an Argentine Ant, Linepithema humile, Semiochemical to Deliver an Acute Toxicant"

_insects, 2018, doi:10.3390/insects9040171_

Round 1

Reviewer 1 Report

General Comments:

The objective of manuscript INSECTS-2018-380146 was to utilize solvent extracts from whole body washes of pupae and dead workers to increase handling of insecticide-treated filter papers, thereby improving treatment efficacy with the aim of reducing insecticide quantity necessary for control. The idea was a good one and worth exploring. The manuscript is well written, presented and analyzed. The main concern I had was the disconnect between pupal extract “attraction,” but failure of the ants to move the treated discs into the nest cell (which the authors admit in the paragraph beginning on line 341.  Also, it is obvious that polar and non-polar components are being removed from the pupae.  Why didn’t the authors try both extracts together?

Specific Comments:

1.      Line 46, change with to at.

2.      Line 53, change to impairment and median lethal time recorded.

3.      Line 110, what year were the ants collected?

4.      Figure 1, move the “d” label closer to the circle in the arena.

5.      Line 162, 3 pumps from a spray bottle is not very precise.  Can you estimate the volume used?

6.      Line 217, The percentages in this paragraph took me a while to figure out.  The table (2) show numbers of ants retrieved, so percentages are not immediately relatable.  Can you change either this text or the table so the data are more easily interpreted/understood. 

7.      Line 341, Why didn’t you try both extracts together?

Author Response

I have printed the comments from Reviewer1 and have agreed with and changed each Specific Comment into the revised manuscript. Reviewer, please see my edits in highlight as they relate to your Specific Comments #1 through #6.

The last comment, #7, is also related to the reviewer's introductory paragraph, where they state "The main concern....Why didn't the authors try both extracts together?" You've identified what are, admittedly, the two "in hindsight we should have done this" points. This clearly would have made the study more complete. The ants may have brought the papers into the nest, but the assay was stopped short of this. And, if we had it to do over we would have washed pupae with one solvent then another and given the ants those papers and watched their response. These are two shortcomings that clearly leave room for future studies should another scientist want to pursue this line of investigation further.

See attached MS to be sure we've addressed your Specific Comments #1-#6.

Reviewer 2 Report

General Comments.- The paper will be of interest to those interested in IPM and lowering the use of insecticide sprays to control ants. I think parts of the abstract and introduction should be re-written. The lack of efficacy of most insecticides and challenges faced by PMPs on residential routes contribute to the excessive use of pesticides to control this ant and generate water runoff issues. It is for these reasons that this research is interesting and potentially valuable.

L. humile is a serious pest and this research could assist in its control in native and ag. situations.

Specific Comments.-

Line 15- L. humile is more than a nuisance pest. It is an important pest in agriculture and especially citrus and grapes where it reduces the effectiveness of biological control programs. In native habitats this species has been able to eliminate many native insects. It is a serious pest.

Line 20- Pesticides in urban waterways is a problem in California. However, it has also been a problem in the northwest. Many urban creeks around the country have been found to contain pesticides. I would imagine that some of that is also attributably to ant control strategies elsewhere.

Line 38- Mayr should read (Mayr).

Line 43.- This is totally incorrect. Most contact insecticides and sprays are ineffective. This has resulted in the over reliance of pyrethroid and fipronil sprays. Klotz and co-authors have numerous papers showing that most spray treatments are not that effective. In the US, there are no effective fipronil baits. The studies cited about fipronil baits were experimental sugar water baits at very low concentrations. There is a fipronil bait “Extinguish” in New Zealand that has been reported as very effective, but it is not registered in the US.

Line 101. This is simply not so. There are no fipronil baits that are effective. Current restrictions on the use of fipronil sprays have dramatically reduced their effectiveness. The sprays do have an ecological drawback when improperly applied. These are the reasons that this research has potential and could be interesting.

Lines 181-186. Not familiar with this analysis. Maybe this could be explained a little better. What does the residual axis -1 to 3 mean? What is the linear predictor?  

Line 289 – Another similar idea was proposed by Choe et al. 2010. Development of virtual bait station to control Argentine ants. J. Econ. Entomol. 103: 1761-1769. The ants picked up the fipronil as they foraged on sugar water. The authors were exploiting the fact that fipronil transfers by contact between ants.

Author Response

I have printed the comments from Reviewer 2 and have agreed with and changed each Specific Comment into the revised manuscript. Reviewer, please see my edits in highlight as they relate to your Specific Comments #1 through #5 of 7 comments. Please check your edits against my revisions to make sure your comments and concerns are addressed appropriately.

Comment #6. As for Specific Comment #6, regarding the statistics, a residual plot (i.e., Figure 3) is used as a visual in order to give the reader confidence that the data are normally distributed and that the stated quantitative technique is applicable.

Comment #7. As for Specific Comment #7, the earliest proposal we could find that this technique could be used against ants was the 1975 paper by Bigley and Vinson.